# Identification and Characterization of Novel SPHINX/BMMF-like DNA Sequences Isolated from Non-Bovine Foods

**DOI:** 10.3390/genes14071307

**Published:** 2023-06-21

**Authors:** Diana Habermann, Martin Klempt, Charles M. A. P. Franz

**Affiliations:** Department of Microbiology and Biotechnology, Max Rubner-Institut, Federal Research Institute for Nutrition and Food, Hermann-Weigmann-Straße 1, 24103 Kiel, Germany; martin.klempt@mri.bund.de (M.K.); charles.franz@mri.bund.de (C.M.A.P.F.)

**Keywords:** circular DNA, SPHINX, BMMF, plasmid, rolling circle replication, theta replication

## Abstract

Sixteen novel circular rep-encoding DNA sequences with high sequence homologies to previously described SPHINX and BMMF sequences were isolated for the first time from non-bovine foods (pork, wild boar, chicken meat, Alaska pollock, pangasius, black tiger shrimp, apple, carrot, and sprouts from alfalfa, radish, and broccoli). The phylogenetic analysis of the full-length circular genomes grouped these together with previously described representatives of SPHINX/BMMF group 1 and 2 sequences (eight in each group). The characterization of genome lengths, genes present, and conserved structures confirmed their relationship to the known SPHINX/BMMF sequences. Further analysis of iteron-like tandem repeats of SPHINX/BMMF group 1-related genomes revealed a correlation with both full-length sequence tree branches as well as Rep protein sequence tree branches and was able to differentiate subtypes of SPHINX/BMMF group 1 members. For the SPHINX/BMMF group 2 members, a distinct grouping of sequences into two clades (A and B) with subgroups could be detected. A deeper investigation of potential functional regions upstream of the *rep* gene of the new SPHINX/BMMF group 2 sequences revealed homologies to the *dso* and *sso* regions of known plasmid groups that replicate via the rolling circle mechanism. Phylogenetic analyses were accomplished by a Rep protein sequence analysis of different ssDNA viruses, pCRESS, and plasmids with the known replication mechanism, as this yielded deeper insights into the relationship of SPHINX/BMMF group 1 and 2 Rep proteins. A clear relation of these proteins to the Rep proteins of plasmids could be confirmed. Interestingly, for SPHINX/BMMF group 2 members, the relationship to rolling circle replication plasmids could also be verified. Furthermore, a relationship of SPHINX/BMMF group 1 Rep proteins to theta-replicating plasmid Reps is discussed.

## 1. Introduction

Circular single-stranded DNA (ssDNA) containing an open reading frame (ORF) encoding a replication initiation protein (Rep protein) that was isolated from eukaryotic cells or tissues could have originated from one of three different sources: eukaryotic, bacterial, or viral. Eukaryotic synthesis of circular ssDNA with a Rep protein encoding ORF has not yet been described, but intermediate single-stranded plasmids of bacterial origin are well known, and metagenomic sequencing has frequently revealed circular ssDNA viruses with Rep encoding ORFs in eukaryotic cells (for review see [1]). Circular single-stranded genomes of CRESS (Circular Rep-Encoding Single-Stranded) DNA viruses replicate by the rolling circle mechanism, which starts with the activity of the virus-encoded Rep protein [1]. On the other hand, circular plasmids of bacteria use various replication strategies, which can generally be grouped into three categories: theta replication, strand displacement replication, and rolling circle replication [2].

In theta replication (TR), the replication of the leading and lagging strand is performed in a coordinated bidirectional way building two independently acting replication forks. The double-stranded DNA (dsDNA) is usually nicked by a plasmid-encoded Rep protein, which binds to highly conserved repetitive DNA elements called iterons. In addition, these iterons are also involved in the control of the *rep* gene expression and the frequency of plasmid replication [3] The nicking of the dsDNA is followed by the unwinding of the DNA by enzymes such as the DnaB helicase, which is often supported by a DnaA protein. The DnaA protein binds to a conserved 9 bp sequence motif called the DnaA-Box, which is located in the (A/T-rich) origin of replication (*ori*). In contrast to the leading strand, the lagging strand is synthesized discontinuously. Under electron microscopy, the early stages of the replication process look like the Greek letter theta (ϴ), from which the name for the replication process was derived [2]. In contrast to this, plasmids replicating by strand displacement (SDR) require three plasmid-encoded proteins (RepA, RepB, RepC), which promote a continuous replication of both strands by using single-strand origins (*sso*) located on each strand [2]. The third mechanism of plasmid replication, rolling circle replication (RCR), is initiated by the generation of a site-specific nick (*dso*) by a plasmid/virus-encoded Rep protein. Nicking of the DNA is followed by the covalent binding of the Rep protein and extension of the 3′ end by a DNA-dependent DNA polymerase III. After completing the single-strand synthesis, transesterification joins the 5′ and the 3′ ends to generate a circular molecule, which is the template for the second-strand synthesis. RCR is the only replication mechanism that does not use primer RNA and is characterized by a single-stranded intermediate [2]. 

Another approach, apart from looking at the replication mechanism, to decide whether circular ssDNA is of viral or bacterial origin is the phylogenetic analysis of its encoded proteins. The Rep proteins, which are essential for replication and therefore highly conserved, are particularly suitable for this purpose [4]. Rep proteins belong to the family of endonucleases, which bind first to a specific nucleotide sequence to initiate replication. A helix-turn-helix (HTH) motif and its winged variant (WH) are usually present in the DNA binding domain. Other motifs like HUH (histidine-unpolar amino acid-histidine) or the Walker motifs are present depending on the replication mechanism. Although the term ‘Rep’ is sometimes used to refer to different functional proteins involved in plasmid or viral replication, here we use the term Rep as an abbreviation for ‘replication initiation protein’. Based on their domain structure, Reps and other proteins can be clustered into Pfam families [5]. The nomenclature of Reps is somewhat confusing, and the letters or numbers used to differentiate Reps are arbitrary. Two Reps with the same name in different plasmids and/or different organisms do also not necessarily indicate that the two Reps are related or even identical.

In 2011, circular ssDNA molecules termed SPHINX were isolated by Laura Manuelidis in murine and hamster cell lines infected with TSE (transmissible encephalopathies), suggesting a prion protein-independent ontogeny of plaque-induced cerebral dysfunction [6]. In 2017, a class of circular ssDNA molecules called BMMF (bovine meat and milk factors) was isolated from bovine milk and serum, as well as patients with multiple sclerosis, and those molecules were very similar to the earlier isolated SPHINX molecules [7]. It was postulated that these molecules are specific for *Bos taurus* and are associated with the development of cancer, occurring decades after ingestion of *B. taurus* milk and meat products [7,8]. The theory of a specific association of these molecules only with samples from *B. taurus* has subsequently been disproven, as several SPHINX/BMMF-like DNA molecules have been detected in other food sources. König et al. were the first to detect these molecules in the milk of water buffaloes [9], sheep, and goats [10]. Khvostov et al. detected SPHINX/BMMF molecules in sheep meat by a real-time PCR approach [11]. Finally, these molecules were also found in a wide range of food products, including vegetables, fruits, fish, seafood, and meat of pork, chicken, and wild boar by PCR amplification [12]. Here, 16 novel full-length SPHINX/BMMF molecules from pork, wild boar, chicken, Alaska pollock, pangasius, black tiger shrimp, apple, carrot, and sprouts from alfalfa, radish, and broccoli are described. Analyses were focused on the comparison with formerly described SPHINX/BMMF sequences. The objective of the study was to detect and characterize full-length sequences that are related to SPHINX/BMMF DNA in non-bovine-derived food. Furthermore, we aimed to determine whether the isolated DNA molecules were of potential viral or prokaryotic origin. Therefore, we searched for conserved components of possible replication mechanisms and investigated the Rep protein homologies in silico. For better readability, the abbreviation S/B instead of SPHINX/BMMF was used. The S/B group numbering was adopted from zur Hausen et al. 2017 [7].

## 2. Materials and Methods

### 2.1. DNA Extraction and PCR

The selection of the food samples was based on the positive results of the study by Pohl et al. 2022 [12]. In this study, pork, wild boar meat, chicken meat, Alaska pollock meat, pangasius meat, black tiger shrimp meat, apple, carrot, and sprouts from alfalfa, radish, and broccoli were investigated. Slices of 150–200 mg of the eatable portion of the food samples were digested for 16–18 h at 37 °C with proteinase K (4 ng/mL) and RNase A (0.1 ng/mL) (GeneON GmbH, Ludwigshafen, Germany) and afterward extracted with the GF-1 Viral Nucleic Acid kit (GeneON GmbH) following the manufacturer’s instructions. Circular DNA molecules present in the isolated DNA were enriched by rapid circle amplification (RCA) using the Phi29 polymerase (New England Biolabs, Heidelberg, Germany) according to the manufacturer’s instructions. Briefly, 50 ng of DNA were denatured for 90 s at 95 °C in the presence of 25 µM exonuclease-resistant random hexamers (Thermo Fisher Scientific Biosciences, St. Leon-Rot, Germany). After cooling down to room temperature and adding 0.25 mM dNTPs and 10 U Phi29 polymerase, the reaction was incubated at 30 °C for 16 h and then terminated by heating to 60 °C for 10 min. After this step, 0.3 to 1 µL of the RCA reaction was used as a template for PCR amplification with group-specific primers [12]. The amplified fragments were separated on a 1% agarose gel by electrophoresis in 1 × TAE buffer. Fragments were cut out of the gel using a scalpel, and DNA was eluted from the gel using the Monarch^TM^ DNA Gel Extraction Kit (New England Biolabs). Purified fragments were cloned into pJet1.2 (Thermo Fisher Scientific Biosciences) or pCR2.1 (Thermo Fisher Scientific Biosciences), ligated, and finally commercially sequenced by Sanger sequencing (LGC Genomics, Berlin, Germany). Alternatively, the PCR amplicons obtained after RCA were directly submitted for sequencing, using PCR primers as sequencing primers. Using the obtained short sequences, new primers, abutting at their 5′ end, were designed to obtain full-length sequences. Primers for each construct and the PCR conditions are summarized in the Appendix A (Appendix A). The DNA fragments generated were ligated, cloned, and sequenced as described above. When necessary, new primers for primer walking were designed. The obtained sequences were assembled using the Geneious Prime Assembling tool (version 2022.2.2). The resulting full-length circular sequences were deposited in the NCBI Genbank database (OQ990774-OQ990789).

### 2.2. ORF Annotation and Structural Analyses

The determined circular sequences were examined for putative protein-coding regions. The ORF prediction was performed using NCBI ORFfinder (https://www.ncbi.nlm.nih.gov/orffinder/) and the Geneious Prime Annotation tool (version 2022.2.2). The parameters were adjusted to find ATG and alternative initiation codons using the standard genetic code (Table 1) as performed in previous works [9,10,13]. The minimal ORF length was set to 75 nt by default. However, only ORFs ≥ 95 amino acids were included in further analyses. The predicted ORFs were first analyzed using the NCBI BLASTp tool (https://blast.ncbi.nlm.nih.gov/Blast.cgi (accessed on 7 December 2022)) and the nonredundant protein sequences database. In a second step, searches for structural homologs based on profile HMMs (Hidden Markov Models) were performed using the MPI bioinformatics toolkit HHpred (https://toolkit.tuebingen.mpg.de/tools/hhpred, (accessed on 7 December 2022) [13]). Predicted ORFs that showed probabilities below 50 % were not annotated. Furthermore, the noncoding regions of the new full-length circular S/B sequences were analyzed. Repeat sequences (tandem repeat, inverted repeat) were detected using the Geneious Prime Emboss tool (version 22.2.2). The conserved 22-bp iteron-like tandem repeat (ITR) sequences were deduced from Clustal Omega Alignments of the ITR regions. Detection of *dso* and *sso* of the new S/B sequences was performed manually. The alignments of the *dso/sso* regions were calculated using Clustal Omega.

### 2.3. Phylogenetic Analyses

In the first step, a comparison of all new full-length circular S/B sequences was performed against the nucleotide collection (nr/nt) database from NCBI using the BLASTn tool (https://blast.ncbi.nlm.nih.gov/Blast.cgi, megablast algorithm (accessed on 3 June 2022)). Phylogenetic analysis of the new S/B full-length sequences was performed using all known S/B group 1 (*n* = 42) and S/B group 2 (*n* = 109, reduced to 48 because of very high sequence similarity) sequences, as well as highly related Blast homologs (*n* = 6 for S/B group 1 and *n* = 4 for S/B group 2). For the phylogenetic analysis of the new S/B Rep proteins, further Rep protein sequences from ssDNA viruses and bacterial plasmids were added. Only putative Rep protein sequences of more than 100 amino acids were included in this analysis. In the case of *Dioscorea rotundata* mt (mitochondrial) DNA TDr96_F1 (LC219423.1) and *Medioppia subpectinata* (OC882703.1), no annotated putative Rep protein sequences were present in the databases; therefore, putative Rep protein sequences were obtained from an annotation of the available nucleotide sequence using the Geneious Prime Annotation tool (version 2022.2.2) according to the abovementioned criteria. All the other sequences were retrieved from the NCBI database and the accession numbers of all used sequences are listed in Appendix A. Nucleic acid sequence alignments were calculated using Clustal Omega (version 1.2.2 [14]), and amino acid alignments were calculated using MAFFT (version 7.490 [15]). Evolutionary analyses were conducted in MEGA X (version 10.2.5 [16]). For tree reconstruction, different models were used depending on the dataset. The analysis of Rep protein phylogeny was performed using a discrete γ distribution (5 categories). The reliability of the inferred trees was tested using the bootstrap method [17]. The maximum likelihood method and Hasegawa-Kishino-Yano model [18] were used to construct the phylogenetic tree of full-length SPHINX/BMMF group 1 sequences to analyze the relationship of iteron-like tandem repeats.

### 2.4. Identification of Putative DnaA-Boxes and oriT

The putative regulatory regions between protein-coding regions were examined for motifs typical for bacterial plasmids and viruses. DnaA-Boxes are DNA motifs that bind DnaA, a protein involved in the initiation of DNA replication in bacteria. It plays a key role in coordinating the replication process and ensuring accurate duplication of the bacterial chromosome or plasmids. After using the search term ‘DnaA box’ within the genetic compartment ‘plasmid’ on the NCBI Nucleotide database (https://www.ncbi.nlm.nih.gov/nuccore/ (accessed on 22 November 2022)), 66 entries were downloaded and the annotated DnaA-Boxes were manually extracted from the sequences. Duplicates were eliminated. The resulting 17 different sequences were manually aligned and a putative DnaA-Box sequence, which includes all possible variations, was identified (Appendix A). The origin of transfer (*ori*T) is a specific DNA sequence found in plasmids and plays a critical role in the conjugation of plasmids. The *ori*T sequence analysis was performed using conserved *ori*T sequences derived from the literature [19]. Furthermore, the S/B sequences were checked for a virus-typical conserved nonanucleotide recognition sequence and stem-loop structure of the viral origin of replication described by Tompkins et al. [4].

## 3. Results

Our previously published results confirmed the existence of circular Rep-encoding DNA molecules in a broad spectrum of non-bovine foods, including meats, fish, seafood, vegetables, and fruits, as well as in animal samples from chicken and pig (saliva, feces), via a targeted PCR amplification and sequencing-based approach [12]. In this work, the full-length sequences of 16 circular Rep-encoding genomes could be successfully assembled and verified. 

### 3.1. Phylogenetic Analysis of New SPHINX/BMMF Molecules

BLASTn comparisons with known S/B sequences confirmed the relatedness of the new circular sequences to S/B- groups 1 and 2 (8 each) (Table 1). The coverage to homologous sequences ranged from 62 to 100% for all of the 16 new sequences based on the best hits obtained in the BLAST search. The highest sequence identities of DNA molecules homologous to S/B group 1 ranged from 82.9 to 100%, and for molecules homologous to S/B group 2, the range was from 82.0 to 95.7% (Table 1). In some cases, high homologies were found in DNA sequences that were not designated as S/B sequences, i.e., bacterial plasmid sequences (Table 1).

To further confirm the relatedness to S/B sequences, a phylogenetic tree using the full-length circular sequences determined in this study, as well as S/B sequences obtained from GenBank, was reconstructed. In total, 116 full-length sequences, including the 16 new circular sequences determined in this study, formerly described S/B group 1 and 2 sequences, as well as other homologous sequences detected via BLASTn, were used to reconstruct a phylogenetic tree using the maximum likelihood method. To investigate whether a relationship exists between the sampling source of the isolated sequences (both references and new) and their position in the phylogenetic tree, we visualized the sampling sources of all tree sequences according to the available information. The inferred phylogenetic tree consisted of two main clusters, representing the sequences of S/B groups 1 (blue) or 2 (green) (Figure 1).

The S/B group 1 cluster was divided into several sub-clusters at different degrees of relatedness. The sequence Pang5 determined in this study belongs to a cluster together with 14 very closely related sequences of S/B group 1 from water buffalo milk (45BAMI.2521, 9BAMI.2522, 22BAMI.2521, 22BAMI.2522, 46BAMI.2522, 8BAMI.2522, 14BAMI.2522, 15BAMI.2522, 17BAMI.2522), healthy cattle blood (HCBI6.252, HCBI6.159), commercial cow’s milk (CMI1.252), sheep milk (Sml3), and goat milk (Gml6). The sequence AlfS1 clusters together with sequence LC19493 from *Dioscorea rodundata* (TDr96-F1) mt DNA and is more distantly related to the latter cluster containing Pang5. The sequence ChiM6 clusters very close together with S/B group 1 sequences from water buffalo milk (24BAMI.2092, 11BAMI.2092, 13BAMI.2092, 21BAMI.2091, 12BAMI.2092), sheep milk (Sml4), and goat milk (Gml2). The sequence Carr3 falls into a cluster with a multiple sclerosis brain isolate (MSBI2.176) and the sequence OC882703 from *Medioppia subpectinata* genomic DNA. More distantly related to Carr3 but in the same cluster are further sequences from goat milk (Gml5), cow’s milk (C1MI.3M.1), and a sequence from healthy cattle blood (HCBI3.108). The sequence Pork8 is also related to the latter cluster but more distantly than the other neighbors of Carr3. The sequences AlaP4 and Appl2 together belong to a cluster with sequences from cow’s milk (C1Mis.3M.1), a plasmid from *Acinetobacter cumulans* (p7_060092), and a plasmid identified from the rat gut microbiome metamobilome (pRGRH0320). The sequence Pork7 is related to a cluster containing four cow’s milk isolates (C1MI.9M.1, C1MI.9M.2, C1MI.15M.1, C1MI.15M.2) and a plasmid from a rat gut microbiome metamobilome (pRGRH0677). 

The S/B group 2 cluster appeared to be divided into two very distinct sub-clusters, S/B group 2 subclusters A and B (Figure 1). The sequences BlTS10, BroS9, RadS9, and AlfS9 fall into subcluster A, and the sequences Pork12, Pang 10, WilB10, and Pang 11 fall into subcluster B. Whereas the sequences BlTS10, BroS9, and Pang10 are more distantly related to the next neighbors, the sequences RadS9, AlfS9, Pork12, WilB10, and Pang10 show a closer relation to neighbors and clusters. The next neighbors of BlTS10 are solely cow’s milk isolates (C2MI.5B.7, C2MI.9B.7, C2MI.13B.1). BroS9, RadS9, and AlfS9 belong to a cluster with sequences from cow’s milk (C2MI.9B.13, C2MI.3A.2, C2MI.7A.8, C2MI.1A.4), water buffalo milk (44BAMI.2371), and a plasmid from *Acinetobacter baumannii* (pTS236). Pang11 shows high relatedness to a sequence from a murine TSE-infected cell line (Sphinx2.36). WilB10 belongs to a cluster with sequences from cow’s milk (C2MI.16B.12, C2MI.9B.5, C2MI.15B.2) as well as sequences from healthy cattle blood (HCBI1.225) and a plasmid from a rat gut microbiome metamobilome (pRGRH0103).

In summary, we could not detect any cluster of sequences from similar sampling sources. On the contrary, we found samples from different bovine and non-bovine sources distributed over all the clusters and sub-clusters for both the S/B group 1 and 2 (Figure 1).

### 3.2. Structural Characterization of New SPHINX/BMMF Sequences

#### 3.2.1. SPHINX/BMMF Group 1

The sizes of the eight S/B group 1-related circular sequences isolated in this study ranged from 1665 to 2522 bp (Figure 2a), and the mol% GC contents ranged from 37.3 to 40.1% (Table 1 and Appendix A). Open reading frame (ORF) prediction was performed using ORFfinder (https://www.ncbi.nlm.nih.gov/orffinder/ (accessed on 7 December 2022)) as described in Materials and Methods. Only one ORF was predicted for the genomes of AlfS1, Carr3, AlaP4, Pork7, Pork8, and Appl2. For the genomes of Pang5 and ChiM6, a second ORF was predicted (Figure 2a). The sizes of the ORFs 1 ranged from 894 to 987 nucleotides (nt), and these putatively encode proteins of 297–328 amino acids (aa). The sizes of ORFs 2 ranged from 303 to 660 nt, and these putatively encode proteins of 100–219 aa. The first three hits of BLASTp analyses are shown in Appendix A. The analyses revealed that ORFs 1 potentially encode a replication initiation protein named RepM (Appendix A). For ORFs 1 of Pang5 (100%) and ChiM6 (99,96%), the highest homologies observed were to previously published S/B group 1 replication proteins (15BAMI.2522, 11BAMI.2092 [9]). The other homologous proteins were encoded by plasmids of different *Enterobacteriaceae* species, such as *Acinetobacter* sp. (AlfS1, Pork7) and *Salmonella enterica* sp. (AlaP4, Appl2) or other Proteobacteria species like *Campylobacter helveticus* (Carr3) [21]. One of the highest protein homologs of ORF 1 from Pork8 was an unnamed protein from *Brugia timori*, a nematode. The protein encoded by ORF 2 of Pang5 was identical (100%) to a multispecies hypothetical protein from an *Acinetobacter* species. The protein encoded by ORF 2 of ChiM6 showed homology (99.96% identity) to the hypothetical protein of Sml4 (S/B group 1, [10] (Appendix A).

To gain more insight into the structural properties of the encoded proteins of these ORFs, an analysis of HMM profiles (Hidden Markov models) was performed using the HHPred tool and the Pfam-A_v35 database (https://toolkit.tuebingen.mpg.de/tools/hhpred (accessed on 7 December 2022)) as described in Materials and Methods. For all ORFs 1, high homologies (97.5 to 100% probability) were found to Rep_3, a replication initiation protein that contains domains of the Pfam family PF01051 according to the HHPred analysis hit. For ORF 2 of Pang5, two motifs with high probability were found. In one of these, a part of the hypothetical protein contained a motif belonging to the DUF6290 family (PF19807), proteins with unknown functions predominately found in bacteria, while in another part, an EVE Domain (PF01878, which is part of the PUA superfamily and is thought to be involved in RNA-binding) was identified. For ORF 2 of ChiM6, a relationship to proteins of the DUF2834 family (PF11196, proteins with unknown function) was detected (Appendix A).

All sequences that were associated with the S/B group 1 contain a conserved inverted sequence motif and iteron-like tandem repeat structures. These features are unique for the S/B group 1 sequences and are distinguishable from S/B group 2 sequences. A deeper analysis of the unique ITR structures from the S/B group 1 sequences was performed. First, the sequences were aligned and checked for ITR-similar consensus sequences. Fourteen different conserved 22-bp ITR sequences (motifs) could be detected (Figure 2b), of which individual sequences also differed slightly in the number of repetitions (ITR3, ITR7, and ITR11). After the alignment of the 22-bp ITR sequences, a phylogenetic tree was reconstructed (Figure 2b). Coloring of the sequences according to the individual ITR motif in the phylogenetic tree of the full-length sequences (Appendix A) as well as truncated Rep protein amino acid sequences (Figure 2c) revealed a consistent grouping of ITR motifs. Six of the new S/B group 1-related sequences (Pang5, ChiM6, Carr3, AlaP4, Appl2, and Pork7) harbor ITR motifs that are closely related to those found in previously published S/B group 1 sequences (Figure 2c). The conserved 22-bp ITR motifs from two of the new S/B group 1 sequences, however, were either unique (Pork8, ITR4), or could only be found in *Dioscorea rotundata* mtDNA (LC219423.1) using the BLASTn homologous search (AlfS1, ITR2) (Figure 2c). Overall, ITR motifs are highly conserved within the sequences and probably essential for the persistence of the DNA molecule.

Upstream of the described ITR region is a region with a low mol% GC-content, where a formerly described 12-bp inverted sequence motif (TAAATGCTTTTA) could be detected in all eight new S/B group 1 sequences (Appendix A). A closer look at intergenic regions upstream of the *rep* genes of all known S/B group 1 sequences, where this 12-bp region is located (Appendix A), revealed very high sequence conservation except for the sequences HCBI4.296 and HCBI5.173. Especially an AT-rich region (approx. 55 bp, average mol% GC content about 23%), occurring upstream of the ITR region, which showed overall sequence identities of 65 to 100%, with 100% sequence identity in highly conserved spots (shown as grey boxes in Appendix A) indicating an essential function of this region. Searching for other functional structures like DnaA-Boxes revealed no match to formerly described DnaA-Boxes from the NCBI nr database. Therefore, a consensus sequence from the published DnaA-Boxes was determined including possible variations (Appendix A). Using this, two identical 8-mer sequences (ATTTTCAT) were detected in the mentioned AT-rich region (Appendix A). The sequences were located in direct association, one on the leading strand and one on the lagging strand, with an overlapping nucleotide (G/C). Furthermore, the new S/B sequences were observed for possible *ori*T regions. A conserved 12bp sequence (TAAGTGCGCCCT) of the *ori*T region of the *Bifidobacterium longum* plasmid pMG1 defined by Park et al. [19] was detected with 100% identity in four of the S/B group sequences—AlfS1, AlaP4, Carr3 as well as Pork8 and ChiM6 with one mismatch (AAAGTGCGCCCT). The conserved sequences were found upstream of the Rep-encoding ORF. Adjacent to the conserved sequence, *ori*T typical inverted repeats were found, but no further transfer or conjugation-associated elements. Deeper investigations of the *ori*T regions were not performed. In addition, the analysis of the AT-rich region of the S/B group 1 sequences for a virus-typical conserved nonanucleotide sequence and stem-loop structure [4] revealed no similarities.

#### 3.2.2. SPHINX/BMMF Group 2

The sizes of the eight new circular S/B group 2-related sequences ranged from 2299 to 2442 bp (Figure 3a) and the mol% GC content from 38.5 to 42%, (Table 1 and Appendix A). ORF prediction as described above revealed three to four hypothetical protein-encoding genes for each of the eight new S/B group 2 sequences. For all predicted ORFs, BLASTp analyses were performed. An ORF encoding a protein that is homologous to a replication initiation protein was found and could be identified in all of the analyzed sequences. These Reps are members of the Pfam family 01446, which is involved in plasmid replication. In most cases, only one Rep protein encoding ORF was predicted (RadS9, BlTS10, Pork12, Pang11, Pang10, WilB10), with sequence identities ranging from 77.63 to 100% (coverage over 95%), and with homologies to different *Acinetobacter* spp. (plasmids) Rep proteins or BMMF2 DNA named sequences (Appendix A). In two cases (AlfS1, BroS9) two (partial) Rep-coding ORFs were detected, both of which showed high homologies to other Rep proteins (90.86 to 96.85% sequence identity) from *Acinetobacter baumannii*. Generally, the sizes of the potential replication proteins ranged from 338–438 aa (211–319 aa for the partial replication proteins) (Appendix A). Furthermore, two small ORFs in close proximity (sometimes overlapping) were predicted, having protein sizes between 111–145 aa or 96–107 aa. For both small ORFs, BLASTp analyses showed homologies to uncharacterized or hypothetical proteins from different *Acinetobacter* spp. or BMMF2 DNA named sequences and, also, to other bacteria like *Klebsiella pneumoniae*, *Neisseria meningitidis,* and *Janibacter anophelis*. Most of the protein homologs were classified as proteins occurring in multiple species (Appendix A).

Furthermore, HHPred analysis, which includes predicted structural properties [14] was performed. For the predicted partial or whole Rep proteins of all analyzed new S/B group 2 sequences, homologies to Rep_1 protein ranging from 96.4 to 99.9% could be found (Appendix A). All of the predicted Rep proteins belonged to the Pfam family PF01446. For the two proteins of the adjacent ORFs, different homologs could be detected. For the larger protein (111–145 aa) high homologies (96.9–97.9%) to the RCR replication regulatory protein RepB (PF10723) could be determined. For all of the ORF 2 proteins (96–112 aa), homologies (98.3–98.9%) to the putative γ DNA binding protein G5P (PF17426) were detected. For the ORF 3 protein (105 aa) of WilB10, a 49.32% homology to a P5-type ATPase cation transporter was found (Appendix A).

To date, no regulatory elements for the S/B group 2 sequences have been described. Therefore, we investigated the intergenic region upstream of the *rep* gene. It is known that Rep_1-like proteins recognize a conserved nicking site in the double-strand origin (*dso*) region. We found that this conserved *dso* nicking site (CTTGATA) in all of the new S/B group 2 sequences. Alignments of the surrounding upstream and downstream sequences to known *dso* regions of *Staphylococcus aureus* plasmid pC194, *Pseudomonas putida* plasmid pPP81, and *Acinetobacter baumannii* plasmid pTS236 revealed a high homology to these bacterial *dso* loci as shown in Figure 3b. Furthermore, we also identified homologies to a conserved sequence (CS-6 site; TAGCGA/T), which is part of known single-strand origins (*sso*) [23]. Alignments of potential *sso* regions, however, also showed a higher variability compared with the variability of the *dso* region (Figure 3b). 

### 3.3. Replication Protein Phylogeny

A maximum likelihood phylogenetic analysis of the Rep proteins encoded by the S/B group 1 and 2 sequences was performed. Several Rep proteins from bacterial plasmids and ssDNA viruses with the known replication mechanism were included (Figure 4), as listed in Appendix A. The Rep protein sequences of the S/B group 1 sequences clustered together and showed a close relationship to Reps of TR plasmids (pIGWZ12, pSC101); RCR plasmids (e.g., pMG1, pNAL8L) could also be found in close proximity. The Rep protein sequences of the S/B group 2 clustered on a branch together with RCR plasmids (e.g., pPP8-1, pC194). The Rep proteins of ssDNA viruses and pCRESS clustered together on a main branch separated from the Rep proteins of the plasmids and the S/B sequences with the exception of pAL5000. Comparisons of the Pfam families of the related plasmid Rep proteins have revealed that pIGWZ12, pSC101, and pNAL8L Reps belong to the family PF01051, which is the same for the Rep proteins of the S/B group 1, while the Rep protein of pMG1(adjacent RCR Rep cluster) belongs to the PF0171 family. Further, the Rep proteins of pC194 and pPP8-1 belong to the family PF01446. They cluster with S/B group 2 Reps, which also belong to PF01446.

## 4. Discussion

It has been postulated that at least the SPHINX/BMMF group 1 molecules cause different types of cancer [24,25]. This theory is based on epidemiological evaluations that the incidence of colon and breast cancer is increased in countries consuming *B. taurus-derived* foods and that S/B molecules are specifically found in *B. taurus* samples [26]. First doubts regarding the latter theory arose after the molecules were detected in the milk of other *Bovidae* species, such as water buffalo, goat, and sheep [9,10]. In addition, it was previously also demonstrated that specific S/B sequences were also detectable in a broad range of food samples of plant and animal origin [12]. However, our previous investigation [12] reported only partial sequences; therefore, in this work, we describe for the first time full-length circular S/B group 1 and 2 sequences found in alfalfa, radish, and broccoli sprouts, as well as in apple, carrot, Alaska pollock, pangasius, black tiger shrimp, chicken, pork, and wild boar. 

As seen in Figure 1, the full-length circular sequences examined were widely distributed among the formerly published S/B group 1 and 2 sequences in the inferred phylogenetic tree. The reconstructed tree clusters of S/B group 1 and S/B group 2 DNA sequences are comparable to previously inferred trees [9,10,27], indicating consistent clustering. The examined sequences also show high sequence homologies to known S/B sequences confirming the phylogenetic relationship. Interestingly, analysis of the sampling source did not show any source-specific branching, indicating that the S/B molecules are broadly distributed in both plants and animals. This disproves the theory that S/B molecules are solely found in taurine-derived samples, supported by the previous findings of König et al. [9,10], Khostovv et al. [11], and Pohl et al. [12].

Structural analyses of described herein S/B sequences also revealed high similarities to previously described bacterial plasmids, as well as to other known S/B sequences. This relationship has already been described for S/B sequences and plasmids, but the question of whether these molecules now belong to viruses or plasmids, or whether they constitute a new or novel ‘intermediate’ class in between these, has not yet been answered. Therefore, we focused our analyses on this question.

For all S/B group 1 and 2 molecules described herein, the main characteristic structures, such as (i) sequence length, (ii) GC-content, and (iii) ORF predictions, were similar to the already known S/B group 1 and 2 sequences (Table 1). The reduction of the ORF size cutoff to 95 amino acids has led to the detection of two further small ORFs in all of the described S/B group 2 sequences. In addition, the two ORFs are also detectable in all S/B group 2 sequences known to date [27]. The mol% GC content of the S/B group 1 and 2 sequences is very low and in the same range as the mol% GC contents of the human genome (41%, [28]) and several bacterial genomes like *Acinetobacter* (39.1%, [28,29]). Because the analysis of the potential replication mechanisms of S/B group 1 and group 2 sequences yields different results, the discussion is presented separately.

In previous analyses some mutual characteristics, which showed similarities to both plasmids as well as to viruses, were proposed for the noncoding region upstream of the *rep* gene start codon of the S/B group 1 sequences. Features such as (i) a conserved palindromic structure, which was proposed as the putative origin of replication (*ori*) with proposed similarities to viral nonanucleotide stem-loop *ori,* and (ii) an iteron-like tandem repeat (ITR) region (3 × 22 bp repeats plus partial repeats) similar to iterons in bacterial plasmids were found [27]. All of these features could also be detected in the new isolated S/B group 1 sequences (Figure 2, Figure 3, Figure 4 and Figure 5). Furthermore, it has been speculated that the evolutionary origin of S/B molecules appears to fall between bacterial plasmids and circular ssDNA viruses. It was suspected that these molecules originated from a unique bacterial plasmid, which was adapted (with viral sequences) to infect and replicate in mammalian cells [27]. In *Genomoviridae*, the origin of replication is characterized by a nonanucleotide sequence enclosed by inverted repeats, which form a hairpin structure and present the nonanucleotide cleavage site to the catalytic tyrosine residue of the Rep protein (motif III) [30]. A relationship with viruses was assumed due to similarities in the secondary structure of an inverted repeat sequence (12 bp) found in S/B group 1 sequences and thought to form a stem-loop structure [27]. However, the proposed 12-bp *ori* sequence is shorter than the viral *ori* sequence (appr. 32 nt) and would confer only a four-nucleotide loop (instead of 10–13 nt), similar to what Tompkins et al. [4] described. Therefore, it is doubtful that the cleavage site would be appropriately accessible for the catalytic recognition site of the *rep* gene. 

For each replication mechanism, special DNA structures are necessary and characteristic. Iteron sequences are proposed to be involved in plasmid replication, especially the iteron-dependent theta replication mechanism and strand displacement mechanism [2]. However, they can also be found on plasmids that replicate via the RCR mechanism [31]. The described replication by strand displacement requires three proteins encoded by the plasmid, and this mechanism is therefore very unlikely for all of the S/B group 1 sequences. Accordingly, the investigated S/B group 1 sequences were checked for structural characteristics typical for RCR or theta replication. The nucleotide sequence of AlfS1, ChiM6, Pork8, Carr3 AlaP4, Appl2, Pang5, and Pork7 (Type 1 molecules) contain (i) an AT-rich region with potential DnaA-Boxes and inverted repeats (Figure 2a), (ii) iteron-like-tandem repeat sequences (Figure 2a), and (iii) a gene encoding a replication protein (Figure 2a). This is characteristic of plasmids that replicate via theta replication. However, plasmids with similar structures and a gene encoding similar Rep proteins (PF01051), such as pNAL8L, pKJ50, or pMG1 [19,31,32] have been shown to produce single-stranded intermediates during plasmid replication, which are indicative of rolling circle replication. On the other hand, several structurally similar plasmids with a Rep protein of the PF01051 family have been shown to replicate via theta replication, i.e., pIGWZ12 [33] and pSC101 [34]. Because of this contradicting information, further investigations are needed to reveal which replication mechanism is used by the S/B group 1 molecules. 

To date, S/B group 2 sequences have not been analyzed for a conserved structure beyond the ORFs detected on the sequence. Therefore, we focused our analyses on the noncoding regions. Rolling circle replication is divided into two parts, (i) the synthesis of the leading strand and (ii) the synthesis of the lagging strand. While the first requires a recognition sequence (*dso*) for the initial cleavage as discussed above, the latter depends on the presence of *sso*. A sequence alignment revealed a presence of a region with high homology to *dso*, found in plasmids of different bacterial species, in all of the new S/B group 2 sequences described in this study (see Figure 3b). This *dso* contains a highly conserved nicking site (Figure 3b), which can be nicked by a conserved tyrosin (Y) residue of the Rep protein. Furthermore, similar *dso* sequences could be found in all published S/B group 2 sequences, as well as the BLASTn homologous sequences used for phylogenetic tree building. Similar conserved *dso* structures were found in plasmids and phages replicating via a rolling circle mechanism, including the *E. coli* phage PhiX174 [35,36,37] and the plasmid pT181 isolated from *Staphylococcus aureus* [38]. In both the *E. coli* phage PhiX174 and the plasmid pT181 replicons, the Rep protein is the replication initiator protein that interacts with a binding site and a nick site located within *dso*. The nick site is cleaved by the catalytic tyrosine of the Rep initiator protein, and the released 3′-OH end acts as the primer for the new DNA strand synthesis. The Rep protein remains covalently bound to the 5′ end of the nicked strand (by a phosphotyrosine bond) until the leading strand replication is terminated and the circular ssDNA is released from the newly replicated dsDNA molecule. The ssDNA molecule is then converted into dsDNA starting from the *sso* region located upstream of the *dso* of replication. So far, several main types of *sso* have been described, of which some contain a consensus sequence of 6 nt (CS-6), which seems to act as a terminator. Plasmids from different bacteria sites similar to CS-6 have been proposed or proven as *sso* [35,36,37,39]. We found a CS-6 site in all of the S/B group 2 sequences as indicated in Figure 3a with adjacent sequences, which presumably form a stem, so that the CS-6 site is presented in the loop. The presence of *sso* upstream of *dso* suggests that S/B molecules of group 2 may be naturally occurring within a biological life form, as the replication machinery requires a host primase (RNA polymerase, DNA polymerase) to complete the synthesis of a new dsDNA molecule (biological synthesis hypothesis). Indeed, as mentioned above, some S/B group 2 homologous molecules have been isolated from bacteria. Most strikingly, a circular *Acinetobacter* phage genome shows similarity to the circular DNA sequences described in this study (see Figure 3b), where the two encoded proteins are proposed to be the coat proteins of the isolated phage [38]. 

## 5. Conclusions

Sixteen new S/B sequences were isolated from a broad spectrum of non-bovine food sources (pork, wild boar, chicken meat, Alaska pollock, pangasius, black tiger shrimp, apple, carrot, and sprouts from alfalfa, radish, and broccoli). All sequences showed high similarity to either S/B group 1 or 2 reference sequences, as shown in phylogenetic and structural analyses. A distinct formation of clusters representing S/B sequences from special food sources could be excluded. Even after thorough in silico analyses, it is still not possible to unequivocally determine whether the described molecules are of bacterial, viral, or ‘intermediate’ origin. Only further biochemical and molecular investigations will finally answer this question. Nevertheless, several indicators point to a bacterial origin: (i) the encoded Rep proteins are closely related to Rep proteins described in plasmids or phages; (ii) the S/B group 2 sequences contain typical bacterial *dso/sso* motifs; (iii) several plasmids with identical structures to S/B group 2 have been described [38,40,41,42]; (iv) the motifs (inverted repeat, ITR, DnaA-Box, *ori*T) identified in S/B group 1 sequences are frequently described as also occurring in plasmids [3]; and finally (v) the highly conserved nonanucleotide *ori* sequence [T/A]A[A/T/G]TTATAC of different Rep protein from *Circoviridae*, *Nanoviridae,* and *Geminiviridae*, which have been determined by Tompkins et al. [4], have not been detected in the sequences described herein (for an overview see Figure 5).

## Figures and Tables

**Figure 1 genes-14-01307-f001:**
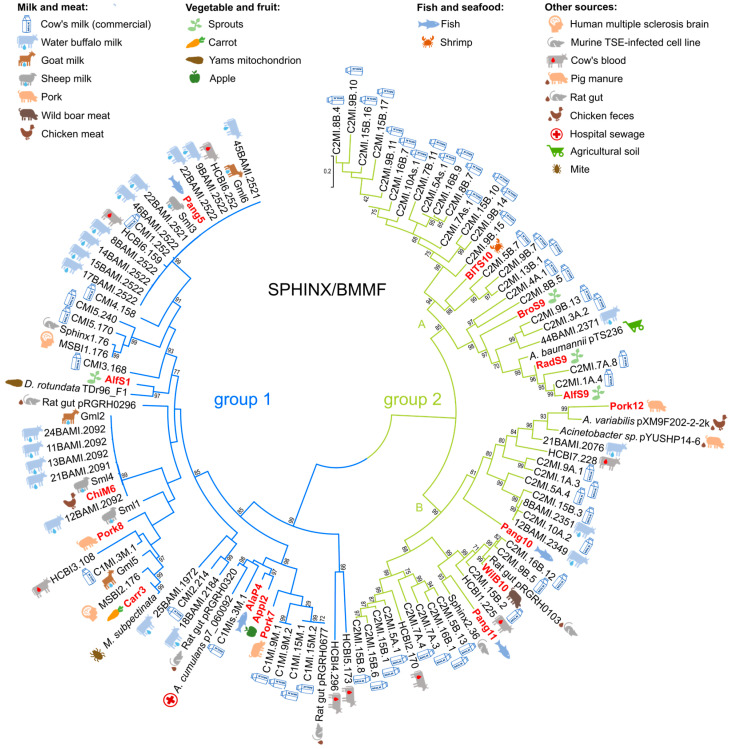
Phylogenetic analysis of 116 SPHINX/BMMF group 1 and group 2 full-length sequences and BLAST homolog sequences. The evolutionary history was inferred by using the maximum likelihood method and the Tamura-Nei model [17,20]. From 250 bootstrap replications, the tree with the highest log likelihood (−139399,47) is shown. Bootstrap values above 70% are shown next to the branches. The tree is drawn to scale, with branch lengths measured in the number of substitutions per site. There was a total of 3496 positions in the final dataset.

**Figure 2 genes-14-01307-f002:**
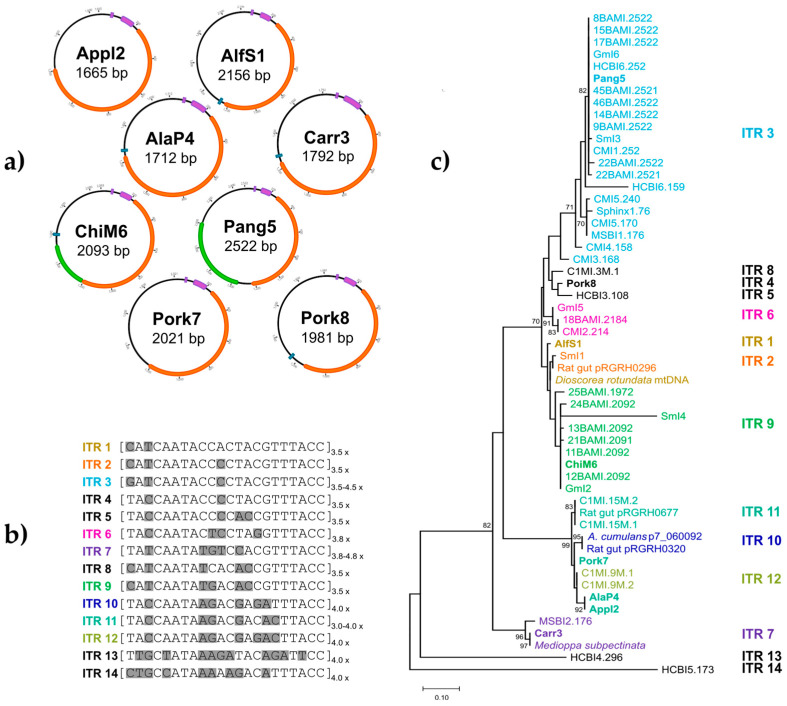
(**a**) Schematic representation of novel circular sequences associated with SPHINX/BMMF group 1. Open reading frames (ORFs) above 95aa (280 nt) are shown. The Rep-encoding ORF is indicated in orange. Further ORFs are indicated in green. The conserved repeat structures (inverted and tandem repeats) are indicated in purple. The detected conserved *ori*T sequence is indicated in turquoise. (**b**) Alignment of 14 differing iteron-like tandem repeat (ITR 1-14) sequences (22 bp) deduced from 56 BMMF group 1 genomes and BLASTn homologs. The aligned 22-bp ITR motif sequences are shown adjacent to the names. All disagreements are highlighted in grey. The number of ITR repetitions is drawn beside the square brackets. (**c**) Phylogenetic analysis of 55 BMMF group 1 Rep protein sequences and BLASTn homologs. The protein sequences were truncated to 143 aa and contain potential ITR recognition regions. The evolutionary history was inferred by using the maximum likelihood method and the Whelan and Goldman model [17,22]. From 250 bootstrap replications, the tree with the highest log likelihood (−1827.08) is shown. Bootstrap values above 70% are shown next to the branches.

**Figure 3 genes-14-01307-f003:**
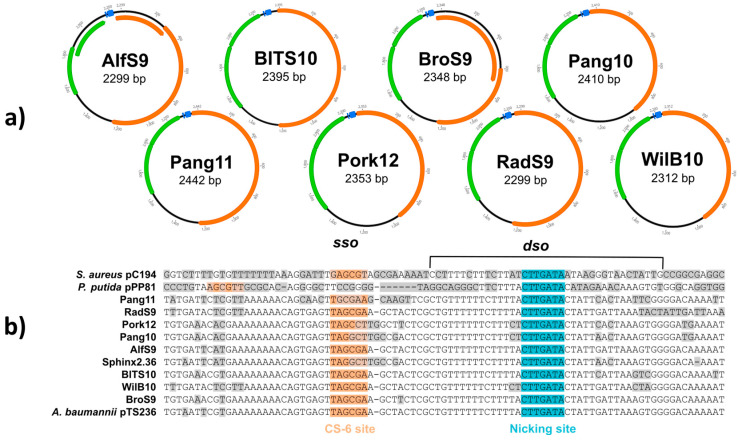
(**a**) Schematic representation of novel circular genomes associated with SPHINX/BMMF group 2. Open reading frames (ORFs) above 95aa (280 nt) are shown. The Rep-encoding ORF is indicated in orange. Further ORFs are indicated in green. The conserved *dso/sso* is indicated in blue; (**b**) Sequence alignments of *dso/sso* regions are grouped by similarity. The differences are shaded in grey. The highly conserved putative *dso* nicking sites and the *sso* (CS-6) sites are colored blue and orange, respectively.

**Figure 4 genes-14-01307-f004:**
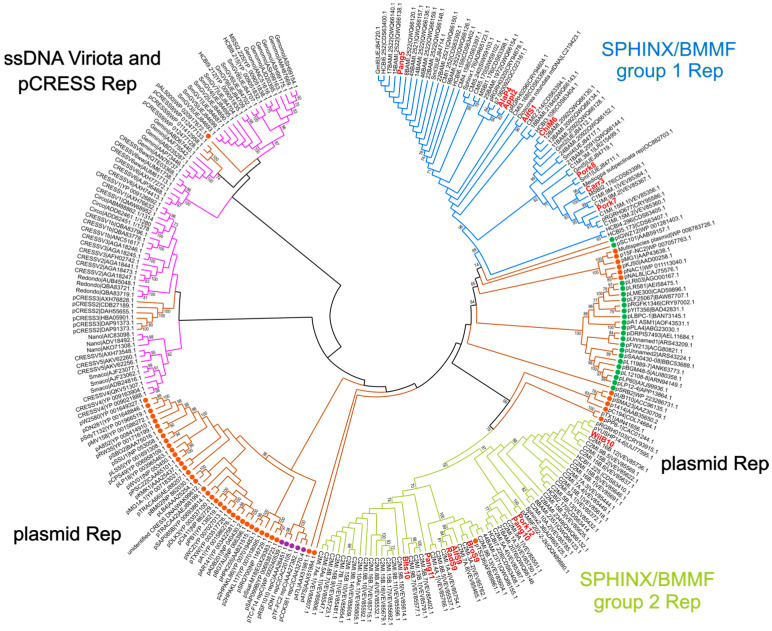
Phylogenetic analysis of Rep protein sequences from ssDNA viruses (pink lines), plasmids (brown lines), SPHINX/BMMF group1 (blue lines), and group 2 (green lines). The evolutionary history was inferred by using the maximum likelihood method and Whelan and Goldman model [18,23]. From 250 bootstrap replications, the tree with the highest log likelihood (−82894.57) is shown. A discrete γ distribution was used to model evolutionary rate differences among sites (5 categories (+G, parameter = 4.2537)). This analysis involved 261 amino acid sequences. There was a total of 899 positions in the final dataset. Bootstrap values above 70% are shown next to the branches. The replication mechanism of several plasmids is indicated next to the labels by a bullet point in orange for RCR, in purple for SDR, and in green for TR.

**Figure 5 genes-14-01307-f005:**
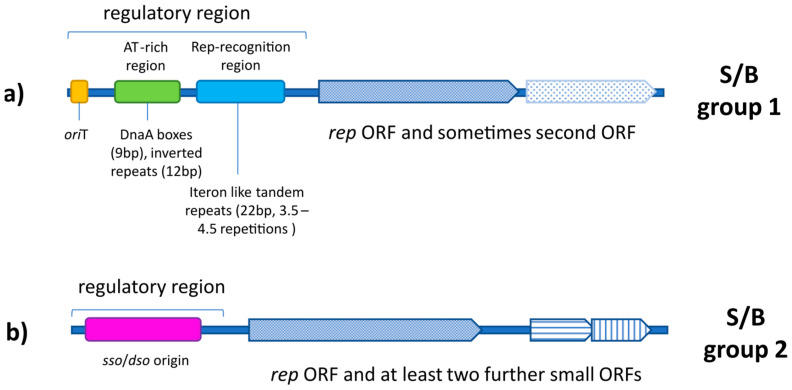
Schematic overview of linearized SPHINX/BMMF group 1 (**a**) and 2 (**b**) sequences and their regulatory region.

**Table 1 genes-14-01307-t001:** BLASTn analyses of complete DNA sequences of circular sequence amplicons from non-bovine food sources generated by SPHINX/BMMF-specific primers (Appendix A). Two or three best homologous hits are shown.

	Name	Source	Query Length (bp)	GC Content (mol%)	Best Hit(s) in BLAST Search	Acc. No. of Hit	Query Coverage (%)	Identity (%)
SPHINX/BMMF group 1	AlfS1	Alfalfa sprouts	2157	37.3	*Dioscorea rotundata* mitochondrial DNA, cultivar TDr96_F1	LC219423	80	98.0
					Uncultured prokaryote from Rat gut metagenome metamobilome, plasmid pRGRH0296	LN852968	72	87.5
					Sphinx1.76-related DNA, replication-competent episomal DNA HCBI6.252	LK931493	73	85.4
	Appl2	Apple	1665	39.5	Uncultured prokaryote from Rat gut metagenome metamobilome, plasmid pRGRH0320	LN852990	98	84.6
					*Acinetobacter cumulans* strain WCHAc060092 plasmid p7_060092, complete sequence	CP035942	86	87.7
					BMMF1 DNA sequence, isolate C1MI.15M.2	LR215495	57	91.3
	Carr3	Carrot	1792	38.4	*Medioppia subpectinata*	OC882703	100	100
					BMMF1 DNA sequence, isolate GmI5	OK148629	94	87.2
	AlaP4	Alaska pollock	1712	40.1	*Acinetobacter cumulans*: WCHAc060092 p7_060092	CP035942	94	88.5
					BMMF1 DNA sequence, isolate C1MI.15M.2	LR215495	73	92.0
	Pang5	Pangasius	2522	36.3	BMMF1 DNA sequence, isolate GmI6	OK148630	100	99.8
					BMMF1 DNA sequence, isolate 15BAMI.2522	MW828665	100	99.7
	ChiM6	Chicken meat	2093	37.8	BMMF1 DNA sequence isolate 11BAMI.2092	MW828660	100	99.6
					BMMF1 DNA sequence isolate 24BAMI.2092	MW828672	100	99.6
	Pork7	Pork	2021	38.0	Uncultured prokaryote from Rat gut metagenome metamobilome, plasmid pRGRH0677	LN853297	81	94.1
					BMMF1 DNA sequence, isolate C1MI.15M.2	LR215495	84	96.9
	Pork8	Pork	1981	37.6	Sphinx1.76-related DNA, replication-competent episomal DNA MSBI1.176	LK931491	62	82.9
					Sphinx1.76-related DNA, replication-competent episomal DNA CMI3.168	LK931489	57	84.1
SPHINX/BMMF group 2	AlfS9	Alfalfa sprout	2299	40.3	BMMF2 DNA sequence, isolate C2MI.1A.4	LR215586	100	95.7
					BMMF2 DNA sequence, isolate C2MI.1A.2	LR215584	100	95.6
	RadS9	Radish sprouts	2299	40.6	*Acinetobacter baumannii* strain DS002 plasmid pTS236, complete sequence	JN872565	99	87.5
					BMMF2 DNA sequence, isolate C2MI.1A.4	LR215586	83	99.2
	BroS9	Broccoli sprouts	2348	40.5	BMMF2 DNA sequence, isolate C2MI.3A.1	LR215587	62	93.3
					BMMF2 DNA sequence, isolate C2MI.3A.2	LR215588	62	93.2
	BlTS10	Black tiger shrimp	2395	39.4	BMMF2 DNA sequence, isolate C2MI.9B.13	LR215550	99	82.9
					BMMF2 DNA sequence, isolate C2MI.9B.12	LR215549	99	82.9
	Pang10	Pangasius	2410	42.0	BMMF2 DNA sequence, isolate C2MI.9A.1	LR215538	63	82.0
					BMMF2 DNA sequence, isolate C2MI.1A.1	LR215583	63	82.0
	Pang11	Pangasius	2442	40.8	TSE-associated circular DNA isolate Sphinx 2.36, complete sequence	HQ444405	71	90.9
					Uncultured prokaryote from Rat gut metagenome metamobilome, plasmid pRGRH0103	LN852793	71	77.9
	WilB10	Wild boar	2312	40.5	BMMF2 DNA sequence, isolate C2MI.15B.2	LR215554	99	93.6
					Uncultured prokaryote from Rat gut metagenome metamobilome, plasmid pRGRH0103	LN852793	99	98.4
	Pork12	Pork	2353	38.5	*Acinetobacter variabilis* strain XM9F202-2 plasmid pXM9F202-2-2k, complete sequence	CP060816	72	92.1
					*Acinetobacter* sp. SH20PTE14 plasmid pYUSHP14-6, complete sequence	CP090073	72	78.0
					BMMF2 DNA sequence isolate 8BAMI.2351, complete sequence	MW828658	68	81.3

## Data Availability

The data presented in this study are openly available at https://www.ncbi.nlm.nih.gov, accession numbers: OQ990774-OQ990789.

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
