# Peer review of "Identification and Characterization of Novel SPHINX/BMMF-like DNA Sequences Isolated from Non-Bovine Foods"

_genes, 2023, doi:10.3390/genes14071307_

Round 1

Reviewer 1 Report

Thank you for sending me this interesting manuscript by Diana Habermann et al.  The authors found and described 16 novel rep-encoding DNA sequences with homologies to SPHINX and BMMF families. They isolated these new DNA elements from non-bovine foods. A deeper investigation of potential functional  regions upstream of the rep gene of the new SPHINX/BMMF group  sequences revealed homologies to dso and sso regions of known plasmid groups which replicate via the rolling circle mechanism. in this work they describe for the first time full-length circular SPHINX/BMMF group 1 and 2 sequences found in alfalfa, radish and broccoli sprouts, as well as in apple, carrot, Alaska pollock, pangasius, black tiger shrimp, chicken, pork and wild boar that was not previously reported.

There are issues that are listed in order, as follows:

1) table 1:  it seems that some words in the table are cut in two parts and some words are cropped (group 2).

2) page 8, line 263: change “Figure 2A” to “Figure 2a”

3) please add more information about Iteron and an additional figure illustrating the iteron-dependent theta replication mechanism and strand displacement mechanism would be more informative.

4) add more details to the conclusion sections.

Moderate editing is needed for a better understanding

Author Response

Comments on the reviews

We are very pleased that the reviewers find our research interesting and worthy of publication. We have taken the reviewers' comments very seriously and have revised the manuscript accordingly. We trust, that we met the mentioned points and critics.

Below are the specific responses in red to the comments.

Reviewer 1

Thank you for sending me this interesting manuscript by Diana Habermann et al.  The authors found and described 16 novel rep-encoding DNA sequences with homologies to SPHINX and BMMF families. They isolated these new DNA elements from non-bovine foods. A deeper investigation of potential functional regions upstream of the rep gene of the new SPHINX/BMMF group sequences revealed homologies to dso and sso regions of known plasmid groups which replicate via the rolling circle mechanism. in this work they describe for the first time full-length circular SPHINX/BMMF group 1 and 2 sequences found in alfalfa, radish and broccoli sprouts, as well as in apple, carrot, Alaska pollock, pangasius, black tiger shrimp, chicken, pork and wild boar that was not previously reported.

 There are issues that are listed in order, as follows:

 1) table 1:  it seems that some words in the table are cut in two parts and some words are cropped (group 2).

Thank you for this comment. We checked the formatting of the table again and fixed the problems. However, it might be that the format of the paper has been changed due to the review process.

2) page 8, line 263: change “Figure 2A” to “Figure 2a”

We changed that.

3) please add more information about Iteron and an additional figure illustrating the iteron-dependent theta replication mechanism and strand displacement mechanism would be more informative.

We added more information on iterons in the introduction, but we feel that an additional figure describing the different replication mechanisms would be too much detail for this paper and is good described in the cited reviews. The mayor objective of this investigation was to determine whether SPHINX/BMMF are more likely of bacterial or of viral origin by distinguishing the possible replication strategy. And we only wanted to give a short overview of the requirements of each replication strategy in the introduction.

4) add more details to the conclusion sections.

We have revised the conclusion to meet the requirements.

Reviewer 2 Report

First of all lots of works has been done for this works. But I have some concerns - 

-Introduction is not easy to follow - especially in first 4 paragraphs it needs to rearrange it and if needed rewrite it.

- citation in introduction is not properly provided.

- wide range of foods indicated for the presence of the fragments in introduction - some examples will be good.

- Objective is needs to be more specific.

-In materials and methods - for DNA extraction how those samples were collected and selected are not provided. Which tissue were used for DNA extraction are not provided. 

- why two plasmids were used is not clear.

-PCR conditions are not provided

-Importance of ORF annotation and Identification putative DnaA boxes and oriT are not properly described.

-Phylogenetic analysis adding importance of viral and bacterial genomes needs more explanation. 

- Result section is very difficult to follow the step by step findings

- All figures are really small, it is really difficult to understand. One figure per page will be good and understandable.

- Discussion needs to be more supportive to the results.

-Conclusion is not straight forward

The paper has lots information but it is difficult to follow and read and it is not reader friendly.

Author Response

Comments on the reviews

We are very pleased that the reviewers find our research interesting and worthy of publication. We have taken the reviewers' comments very seriously and have revised the manuscript accordingly. We trust, that we met the mentioned points and critics.

Below are the specific responses in red to the comments.

Reviewer 2

First of all lots of works has been done for this works. But I have some concerns - 

-Introduction is not easy to follow - especially in first 4 paragraphs it needs to rearrange it and if needed rewrite it.

The authors are grateful for this advice. We have rewritten the entire introduction to make it more compelling and shorter. We think that it is now an appropriate introduction to the manuscript.

- citation in introduction is not properly provided.

In the course of revising the introduction, we checked all citations and ensured that all statements are documented by citation.

- wide range of foods indicated for the presence of the fragments in introduction - some examples will be good.

To take up this suggestion, we have listed an example of foods in which SPHINX/BMMF sequences have been detected.

- Objective is needs to be more specific.

The objectives of the investigations have also been clarified in the course of rewriting the introduction.

-In materials and methods - for DNA extraction how those samples were collected and selected are not provided. Which tissue were used for DNA extraction are not provided.

We added the requested information in this section. 

- why two plasmids were used is not clear.

We used two plasmids due to the availability of the plasmids in the lab. Both plasmids are useful to clone DNA. Cloning of the fragments have only be done for sequencing purposes.

-PCR conditions are not provided

We added a sentence in the manuscript and added the conditions in Supplement S1.

-Importance of ORF annotation and Identification putative DnaA boxes and oriT are not properly described.

The characterization of the SPHINX/BMMF sequences were performed according previous studies. In addition, we decided to have a closer look to the families of the rep proteins to support the Rep protein phylogenetic analysis.

-Phylogenetic analysis adding importance of viral and bacterial genomes needs more explanation. 

More information has now been added on why phylogenetic analyses can help to determine whether certain DNA structures are bacterial or viral in origin.

- Result section is very difficult to follow the step by step findings

We agree, that these are complicated issues and not very easy to understand, but we also need to present the results in a way that is correct and unambiguous. Due to the changes in introduction and discussion, we think that the results are now easier to read.

- All figures are really small, it is really difficult to understand. One figure per page will be good and understandable.

We agree with that. It has now been arranged in such a way that only one image per page can be seen. However, we used a preformatted document and tried to enlarge the figures as much as possible. Furthermore, the figures are vector graphics and they can be scaled up, so that the readability is significantly improved. In addition, we used color codes to make the figures easier to understand.

- Discussion needs to be more supportive to the results.

The discussion is now revised to be more directly supported by the results to make the link to the results clearer. Furthermore, we added information of the evaluated viral origin sequences in material and methods and results, to give a better context for the argumentation in the discussion.

-Conclusion is not straight forward

We changed the conclusion according to the requested changes.